# Unveiling gender inequality in the US: Testing validity of a state-level measure of gender inequality and its relationship with feminist online collective action on Twitter

**Bruno Gabriel Salvador Casara**[1]*, **Alice Lucarini**[2], **Eric D. Knowles**[3], **Caterina Suitner**[4]

1 Science Division, New York University Abu Dhabi, Abu Dhabi, United Arab Emirates, 2 Faculty of Medicine, University of Modena and Reggio Emilia, Reggio Emilia, Italy, 3 Department of Psychology, New York University, New York, NY, United States of America, 4 Department of Developmental Psychology and Socialization, University of Padova, Padova, Italy

* bgc5630@nyu.edu

**Data Availability Statement:** Public data are available on OSF at: https://osf.io/azs7e/?view_only=c0878592e8eb49b28edb8fd3a7287e2a Part

## Abstract

The Gender Inequality Index is a country-level measure of gender inequality based on women's levels of reproductive health, social and political empowerment, and labor-market representation. In two studies, we tested the validity of the GII-S, a state-level measure of gender inequality in the USA. In Study 1, the GII-S was associated with objective and subjective measures of wellness among women, including life satisfaction, financial well-being, and perceptions of safety. GII-S was not associated with the Gini coefficient, a well-established measure of economic inequality, suggesting that gender and economic disparities represent distinct aspects of social inequality. Study 2 tested the link between GII-S scores and collective action—specifically, participation in the #MeToo movement promoting awareness of sexual harassment and violence against women. Analysis of geo-localized messages on the Twitter social media platform reveals that higher GII-S scores were associated with fewer tweets containing the #MeToo hashtag. Moreover, GII-S was associated with state-level political orientation: the more conservative a state, the higher its level of gender inequality. Results are discussed in terms of possible socio-cognitive processes underpinning the association between gender inequality and sensitivity to violence against women.

## Introduction

Gender inequality is a pervasive global problem that affects rich and poor countries alike [1]. A 2019 Oxfam report [2] showed that nearly two-thirds of the world's 781 million illiterate adults are women, a proportion that has remained largely unchanged for two decades. Moreover, 153 countries' legal systems still discriminate economically against women, including 18 in which husbands can legally prevent their wives from working. Unfortunately, the situation has only worsened since the emergence of the novel coronavirus, with women 24% more likely than

of the data cannot be shared publicly because are owned by a third party (Gallup), and authors do not have permission to share the data. Description of the Dataset and Third-Party Source: The dataset utilized in this study is derived from Gallup's US Dailies, which comprises daily surveys conducted across the United States. These surveys capture a wide range of demographic, social, and economic variables, providing valuable insights into public opinion and behavioral trends. Gallup, an established research organization known for its rigorous polling methodology, is the third-party source of the data. Verification of Permission to Use the Dataset: Our access to the Gallup's US Dailies data was made possible through an institutional agreement between Gallup and our authors' institution (New York University). This agreement authorizes the use of the data for academic and research purposes. While we, the authors, do not own the data, our use complies with the terms set out in the institutional license agreement with Gallup. Special Privileges in Accessing the Data: The access to the Gallup's US Dailies dataset was facilitated by an existing agreement between our institution and Gallup, which may not be universally available to all researchers. No additional special privileges were granted to us beyond what is stipulated in the institutional agreement. Researchers affiliated with other institutions are encouraged to consult their own institutions or directly negotiate with Gallup for access. Contact Information for Data Access: Researchers interested in accessing the Gallup's US Dailies data can initiate their inquiries by contacting Gallup directly. https://www.gallup.com/analytics/213701/us-daily-tracking.aspx.

**Funding:** The author(s) received no specific funding for this work.

**Competing interests:** The authors have declared that no competing interests exist.

men to have lost their jobs permanently during the Covid-19 pandemic [3]. The World Economic Forum's latest report [4] indicates that gender parity remains a distant goal, such that it will take an estimated 132 years to reach equality between genders at the current pace of progress. The fact that gender inequality is still pervasive—even in countries that supposedly lead the world—was confirmed by the agenda for the 2021 meeting of the Group of Twenty (G20) nations, which emphasized the importance of safeguarding women's rights and addressing gender inequalities [5].

Gender inequality impacts numerous life domains, including health [6–8], labor-market participation [9], academia [10, 11], and politics [12]. Women are systematically more likely than men to be victims of sexual harassment [13], are paid consistently less than men [14–16], and are less likely to be hired than men, especially in historically-male professions [17].

Gender inequality has implications, not only for the women who bear the brunt of disadvantage, but for society as a whole. Countries in which women are systematically disadvantaged experience relatively low rates of economic growth [18–20], and the exclusion of women from education and the workplace is detrimental to the growth and development of entire communities [21, 22]. Indeed, gender inequality is associated with poor educational outcomes [23, 24] and government corruption [25].

Fighting gender inequality requires measuring it, and specific instruments have been developed to assess disparities between men and women. One of the most important measures of gender inequality is the Gender Inequality Index (GII; [26]), a measure developed to assess gender inequality at country level, but never used for within-country comparisons. Although country-level comparisons are relevant for providing a broader perspective on social phenomena across a range of cultures and contexts, they often face challenges due to varying data collection methods and cultural differences, introducing methodological inconsistencies (e.g., "qualitative" ideological differences, [27]) and making strict quantitative analysis difficult. Conversely, within-country comparisons, such as between regions within the same nation, can offer more reliable insights into structural phenomena such as gender inequality, by avoiding these issues, benefiting from a unified political system and standardized data practices.

The United States offers a unique setting for comparing gender inequality within a country. Despite shared cultural elements and uniform data collection by federal agencies, US states exhibit significant diversity in demographics, laws, and attitudes on issues linked to gender inequality themes. Importantly, a within-country comparison across US states also provides an opportunity to focus on a specific feminist movement that was born in the country and rapidly became one of the biggest inspirations for collective action in the last decade—namely, the #MeToo movement.

While massively popular, the movement has encountered a great deal of protest and criticism, especially among those who endorse a more traditional worldview regarding gender roles. Indeed, studies indicate that sexist beliefs and perceptions of #MeToo's effects significantly affect individuals' willingness to support the movement [28]. Moreover, Kunst and colleagues [29] reported that men's negative perceptions of #MeToo were linked to higher levels of hostile sexism and lower feminist identification, highlighting ideological rather than gender-based divisions. Additionally, internalized sexualization among women in Italy was found to negatively affects attitudes toward #MeToo and suggests that such internalization may reinforce ideologies that hinder the movement's goals [30].

Importantly, further evidence highlights that ideologies are strictly related to how people react to gender inequality. For example, people high in patriarchal gender-role beliefs tend to rationalize gender inequality as fair, appropriate, and inevitable [31]. Moreover, conservative political ideologies, broadly speaking, serve to "palliate" (i.e., reduce aversion to) the experience of inequality by portraying the world as a fair place where people get what they deserve

[32]. Not surprisingly, then, gender traditionalists and political conservatives tend to tolerate sexual harassment and violence more than do liberals and gender progressives [33, 34]. Consistently, people who endorse right-wing political ideology are also more skeptical toward women who report sexual harassment after a delay (vs. immediately; [35]).

Because #MeToo is so closely tied to a specific form of communication, namely the use of a hashtag on the Twitter platform, the analysis of #MeToo tweets presents a unique opportunity to investigate feminist activism and its relationship to gender inequality and political ideology. More generally, tweets have proven a reliable proxy for a number of social-psychological phenomena, such as personality [36], crime rates [37], and election outcomes [38].

As previously anticipated, the Gender Inequality Index (GII) is one of the most prominent measures of gender inequality and, while it's not singular in its category, we chose it for several reasons. First, the GII is favored by the United Nations Development Programme (UNDP), arguably the international entity most active in efforts to increase gender equality. The UNDP uses the GII to assess the efficacy of interventions designed to reduce country-level disparities between men and women. Validating a state-level version of the GII would provide directly comparable data within the United States.

Another advantage of the GII is that it is multifaceted. The GII accounts for inequality along three dimensions—health, labor, and politics—one or more of which is absent from other measures (e.g., the Basic Index of Gender Inequality, or BIGI; [39]). Women's sociopolitical empowerment, health, and treatment in the workplace are inextricably linked. For instance, to the extent that women lack representation in government, or the educational opportunities necessary to advance in government or industry, their efficacy as advocates and agents of change in other domains of gender inequality will be limited. In line with this claim, evidence suggests that all-male committees are more affected by gender bias in their decision-making than are mixed-gender committees [40]. Moreover, the gender gap in terms of women's political empowerment is currently larger than the gap in other domains [41]. Consistently with these data, ensuring that women are represented at all levels of government is one of the aims of UNDP 2030 agenda (sub-goal 5.5).

## The present research

Across two studies, the present research employs the Gender Inequality Index (GII) and establishes its utility in assessing state-level differences in gender disparities within the US.

Study 1 employed population-based data retrieved from several platforms for big data storage, to create and test the validity of the state-level Gender Inequality Index (GII-S), namely an adaptation of the GII allowing within-country comparisons, which assesses State-level gender inequality across US states. In Study 2, we employed Twitter-based data to validate the GII-S externally, by examining its correlation with the prevalence of #MeToo tweets as a marker of feminist collective action and to explore the interplay between gender inequality, political ideology, and such collective actions across states.

## Study 1

### Method

**Calculation of GII-S scores.** GII-S scores were calculated according to the formula used by [42] to create the country-level GII. Fig 1 displays the components of this formula, specifies how individual measures correspond to the different dimensions of gender inequality, and cites the data sources for each indicator. The inequality data are from 2016, the most recent year information for all indicators was available.

## Gender Inequality Index

| Reproductive health | Empowerment | Labour Market |
|---|---|---|
| 1. Maternal mortality rate (CDC, 2016)<br>2. Adolescent mothers rate (Martin, Hamilton, Osterman, Driscoll and Drake, 2018) | 1. Population with secondary education (Center for American Women and Politics, 2016)<br>2. Female and male shares of parliamentary seats (Gallup, 2019) | 1. Female and male Labour force participation rates (United States Census Bureau, 2016) |

**Fig 1. Dimensions and indicators of the Gender Inequality Index.**

GII-S scores were calculated for 47 out of 50 US. states. Alaska, Hawaii, and Vermont were excluded from the analysis because data relevant to the sub-dimensions of inequality could not be located. In line with Bulmer (1979), the distribution of GII-S scores was approximately symmetrical (skewness = -.034) and normal (Shapiro-Wilk $W = .98$, $p = .68$). The GII-S distribution's central tendency ($M = .28$, $SD = .05$, $Mdn = .28$) indicates a 28% deviation from parity at the expenses of women, suggesting that the US states are substantially gender-unequal overall (Fig 2).

**Convergent and discriminant validity measures.** In order to test the GII-S's convergent validity, we assessed the following state-level indicators retrieved from Gallup US Dailies for 2016 [43]: women's financial well-being (a component of well-being that includes managing economic life to reduce stress and increase security), women's health problems (e.g., "Do you have any health problems that prevent you from doing any of the things people your age

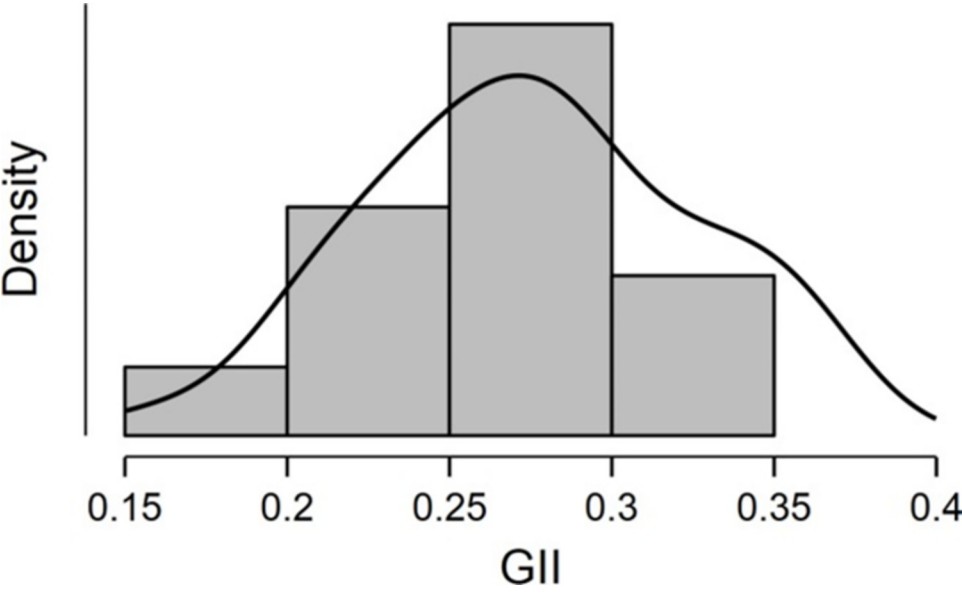

**Fig 2. Density distribution for US. states' GII scores.**

normally can do?"), women's health rating (e.g., "Would you say your own health, in general, is . . .?"), women's perception of safety (e.g., "You always feel safe and secure") and women's life satisfaction (e.g., "I like what I do every day"). For comparison, we also examined these outcome measures for men. The GII-S's discriminant validity was tested against a state-level indicator of economic inequality, the Gini coefficient, for 2016 (One-year estimates, [44]).

**Statistical analyses.** We employed Pearson correlation to examine the relationships between GII-S scores and the other state-level indicators. Pearson correlation was chosen for its suitability in measuring linear relationships between continuous variables, crucial for assessing the predictive power of GII-S scores.

## Results

Results showed that GII-S scores correlated negatively with the per capita GDP, consistent with previous studies finding a negative relationship between gender inequality and economic growth. For both women and men, GII-S scores were negatively associated with objective health (number of health problems), subjective health (health self-rating), and financial well-being, although the association between GII-S scores and financial well-being was stronger for women than men. Consistent with previous literature [45], GII-S scores were also associated with life satisfaction and perceived safety for women. Finally, while GII-S was associated with several well-being indicators, we did not find a statistically significant association between GII and GINI scores. See Table 1 for bivariate correlations between all measures.

## Study 2

### Method

**Tweets collection.** Using R statistical software [46], along with the Rtweet [47] and Revgeo [48] packages, we collected tweets sent in the United States from January 18, 2019, to February 20, 2019. This specific timeframe was selected as matches the peak period of activity

**Table 1. Association between state-level GII and validity measures.**

| | | 1 | 2 | 3 | 4 | 5 | 6 | 7 | 8 |
|---|---|---|---|---|---|---|---|---|---|
| 1. GII score | | — | | | | | | | |
| 2. Gini coefficient | | .043 | — | | | | | | |
| 3. GDP per capita | | -.428*** | .406** | — | | | | | |
| 4. Financial Well-being | F | **-.642***** | -.443** | .493*** | — | | | | |
| | M | **-.397**** | -.283 | .437** | — | | | | |
| 5. Health problems | F | .475** | .003 | -.619*** | -.656*** | — | | | |
| | M | .495*** | .057 | -.702*** | -.522*** | — | | | |
| 6. Health rating | F | -.637*** | -.380* | .552*** | .847*** | -.704*** | — | | |
| | M | -.668*** | -.067 | .650*** | .417** | -.704*** | — | | |
| 7. Perceived safety | F | **-.308*** | -.508*** | -.077 | .501*** | -.147 | .475** | — | |
| | M | **-.181** | -.502*** | -.225 | .161 | .144 | .167 | — | |
| 8. Life satisfaction | F | **-.311*** | -.477** | .158 | .694*** | -.554*** | .596*** | .595*** | — |
| | M | **.051** | .068 | .046 | .282 | -.418** | .175 | .062 | — |

Correlation coefficients

\* p < .05

\*\* p < .01

\*\*\* p < .001

Correlations coefficients that differ at *p* < .05 between males and females, only for the GII scores, are marked in bold.

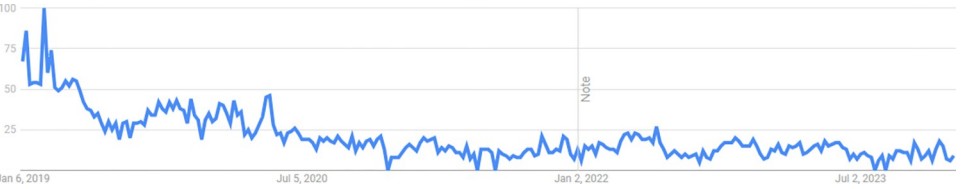

**Fig 3. Google search volumes from January 6<sup>th</sup> 2019 to December 31th 2019.**

for the movement, based on Google Trends data, over the past five years (Fig 3), capturing the height of public engagement and discourse on the Google platform.

Tweets were divided in two groups: those containing the hashtag #MeToo ("#MeToo tweets"; $N$ = 3,191) and tweets randomly sampled from the universe of all US. tweets during the specified time window ("general tweets"; $N$ = 20,000).

The political orientation of each state was determined by analyzing state-level aggregated data from the Gallup US Dailies (Gallup, 2019). Gallup conducts surveys with representative samples of the U.S. population, asking respondents to identify their political beliefs using categories such as *very liberal*, *liberal*, *moderate*, *conservative*, and *very conservative*. To construct a scale estimating the overall political orientation of each state, we subtracted the frequency of *liberal* or *very liberal* responses from the frequency of *conservative* or *very conservative* responses and rescaled the result to range from -1 (a state with only liberals) to 1 (a state with only conservatives).

**Statistical analyses.** In line with the approach used in Study 1, Study 2 employed partialized Pearson correlation to examine the links between GII-S scores, #MeToo tweet prevalence, and political orientation. Once again, Pearson correlation was employed due to its appropriateness for evaluating linear associations among continuous variables. Moreover, in Study 2 the analysis included adjustments for potential confounding factors, such as general tweet frequency, median women's age by state, economic inequality measured with the GINI index, and GDP per capita to ensure the accuracy of the findings. Furthermore, we employed the Bayes Factor, a statistical measure used to evaluate the strength of evidence in favor of one statistical model over another, to test different models to compare the predictive strength of GII-S scores against other models. Specifically, we calculated Bayes factors, using the software JASP [49], for three linear models using #MeToo tweet frequency as dependent variable. The first (null) model included the state-level frequency of general tweets, which should trivially be associated with the use of any given hashtag (including #MeToo), state-level median women age, GINI, and GDP per capita. In the second (GII) model, we added state-level GII in the regression model with #MeToo frequency as dependent variable. Finally, in the third (political) model, we added state-level political orientation, instead of GII, in the regression model with #MeToo frequency as dependent variable.

## Results

In Study 2, we found support the relationship between state-level GII and frequency of #MeToo tweets ($r$ = -.45, $p$ = .003) even after adjusting for general tweets, women age, GINI, and GDP per capita. Differently, we did not find support for the relationship between state-level political orientation and frequency of #MeToo tweets ($r$ = -.24, $p$ = .14). Finally, we found a statistically significant correlation between state-level GII and political orientation ($r$ = .75, $p$ < .001).

According to the Bayes Factors analysis, the GII model was superior to both the null and political models. In particular, the observed data pattern is approximately twenty-eight times

**Table 2. Comparison of models in Study 2.**

| Model | BF$_{10}$ | R$^2$ |
|-------|-----------|-------|
| Null | 1.000 | 0.839 |
| GII | 27.000 | 0.880 |
| Political | 0.448 | 0.848 |

BF = Bayes factor. All models are adjusted for general tweet rate, GINI, GDP per capita, and median women age by State.

more likely to occur under GII model than null model, which can be interpreted as strong evidence [50] in favor of the model with GII. Results of the model comparison are reported in Table 2.

It is important to note that the addition of GII improved upon a model (i.e., the null model) that already explained a large amount of variance. That is, although states that originated more tweets also produced more instances of the #MeToo hashtag, state-level GII explained additional variance in the prevalence #MeToo across states.

## Conclusions

In two studies, the present research develops the GII-S and demonstrates its effectiveness in evaluating variations in gender disparities among US states. Importantly, the US is a federal republic consisting of 50 states bound together in a political union and is particularly well-suited to within-country comparisons with respect to gender inequality. On the one side, these states share important cultural similarities: a common dominant language (English), currency (the US dollar), and bipartisan political framework (Democrats and Republicans). On the other side, the US are highly heterogeneous in terms of demographics and are afforded substantial sovereignty in enacting their own laws, drawing their own legislative districts, and setting their own economic priorities. Moreover, US states differ profoundly in their populations' attitudes on many issues, such as abortion [51], party politics [52], and immigration [53]—all of which, notably, are theoretically related to themes of gender inequality. In sum, state-level *heterogeneity* along dimensions relevant to inequality, coupled with federal *homogeneity* of data collection and management procedures, make the US the ideal context in which to measure within-country differences in gender inequality.

Results from Study 1 highlight that GII scores across US states are coherently associated with state-level well-being indicators. First, in line with previous literature, gender inequality is associated with health outcomes for both males and females. Indeed, public health can be negatively impacted by gender inequality for several reasons. Specifically, the mental health of mothers, which is likely to suffer in highly unequal settings, is an important predictor of male and female children's mental health [54–56]. Moreover, because infectious diseases are more prevalent among individuals low in socioeconomic status, who tend disproportionately to be female, women may tend to spread infections to community members of both sexes [57]. Finally, women are more likely to contract, and subsequently spread, sexually-transmitted diseases in settings where gender inequality leads to higher rates of sexual violence [58, 59].

Findings of Study 1 also suggest that gender inequality is negatively associated with financial well-being for both males and females, although this association is stronger for women. This coheres with previous literature showing that gender inequality is negatively associated with economic growth, leading to less financial well-being for all members of the society [18]. Moreover, a lack of job opportunities for women likely places a burden also on male earners, who face the added stress of supporting a family without the benefit of spousal earnings [60].

Finally, gender inequality is specifically associated with women's perceptions of safety and life satisfaction. It is worth noting that, in our data, women's safety perceptions are linked to their financial well-being, while this association is not statistically significant for men. In contexts of high gender inequality, women tend to be economically dependent on men and therefore have less control over their lives. This can make them more vulnerable to domestic violence or abuse [61]. Thus, gender inequality is robustly linked with poorer life satisfaction among women.

Our findings indicate that GII-S scores are not significantly correlated with economic inequality, as measured by the state-level Gini coefficient. Our findings align with the idea that economic inequality and gender inequality are distinct phenomena. For example, reducing disparities between men and women does not automatically imply that economic differences between classes will follow the same trend. Moreover, coherently with our results, from the 1990 Country-level gender inequality measured with the GII is decreasing in the World and in many countries, USA included [62]. Differently, economic inequality appears to be more stable over time and it increased in the USA during the same time period [63]. However, it is important to notice that there are social issues representing both facets of gender and economic inequality, such as the gender wage gap. Furthermore, our data shows that the Gini coefficient is associated with women's financial well-being (but not men's), and there is previous research reporting a link between economic and gender inequality [64]. Additionally, the distinctiveness between the two issues may be particularly salient because of the specific aspects of gender inequality that the GII-S measures, which do not directly account for gender economic inequality. Therefore, we advise caution in interpreting the lack of a significant association between these two measures.

In sum, Study 1 provides evidence for the convergent and discriminant validity of the state-level GII (GII-S), with GII-S scores associated women's objective and subjective well-being at the state level but not state-level economic inequality.

Consistently with evidence showing that tweets have proven a reliable proxy for a number of social-psychological phenomena [36–38], Study 2 employs the analysis of tweets to investigate feminist online activism and its relationship to gender inequality and political ideology. Specifically Study 2 provides strong evidence that gender inequality is associated with lower commitment to feminist online collective action, such as spreading messages on Twitter related to the #MeToo, an international movement that, after starting in the US, quickly spread worldwide.

Results showed that states with high levels of gender inequality tended to produce fewer #MeToo tweets than did states low in gender inequality. One possible interpretation of these findings can be framed in the theoretical account of the System Justification Theory, in line with our preregistered hypothesis (SJT; [65], link to the pre-registration: https://osf.io/reqxm). According to SJT, people in macro-level contexts characterized by a long history of patriarchy and adherence to traditional sex roles will be motivated to justify gender inequality—a process that likely reduces interest in feminist collective action. In this way, inequality's very existence may further resistance to change. However, it is important to highlight that, while our results are consistent with an interpretation based on System Justification Theory (SJT), they do not conclusively prove it. Other interpretations are also plausible. For instance, according to the Social Identity Model of System Attitudes (SIMSA; [66]), the association between higher GII scores and lower frequency of #MeToo hashtags might be due to accuracy motives. These motives could lead individuals to passively perceive and acknowledge the status quo without actively defending or maintaining the system. Additionally, as the GII measures objective structural inequalities between men and women, it is possible that in States with higher levels of gender inequality, there is a lack of resources and opportunities to participate in the

#MeToo movement. Resource scarcity may also represent an obstacle in the expression of important psychological antecedents of collective action, such as feelings of empowerment and group efficacy beliefs [67]. It is also worth considering that our measure of collective action engagement was specific to the #MeToo hashtags. Future research should examine other forms of collective action, such as participation in demonstrations and strikes. By doing so, researchers might gain a more comprehensive understanding of the dynamics at play, while also providing a more direct test of their possible explanation.

Moreover, our analysis reveals that economic factors like GINI and GDP per capita do not diminish the association between the Gender Inequality Index (GII) and the frequency of #MeToo tweets. This finding suggests that the fight to gender inequality, as captured through the lens of the #MeToo movement, transcends economic disparities at the national level. It indicates that discussions surrounding feminism and women's rights, as highlighted by the #MeToo movement, are prevalent and resonate across states that vary in terms of economic inequality and prosperity.

Finally, in Study 2 we did not find evidence for a relationship between state-level political orientation and #MeToo tweet frequency. It is possible that political orientation has a more nuanced relationship with collective action for women's rights, and conservative variants of feminism are likely intertwined with the political values of conservative women [68]. Future studies are therefore needed to further explore the interplay of individual-level variables (e.g., gender, political orientation, and feminist activism) with structural variables (e.g., state-level gender inequality, political ideology). Conversely, a positive significant correlation between state-level GII and political orientation did emerge in Study 2. This result is consistent with research finding an association between conservative ideology and legitimization of inequality toward women: for instance, people high in patriarchal gender-role beliefs tend to rationalize gender inequality as fair, appropriate, and inevitable [31] and people who endorse right-wing political ideology, gender traditionalists, and political conservatives tend to tolerate sexual harassment and violence more than do liberals and gender progressives [33–35]. Therefore, from a certain point of view, our results concerning structural indicators of gender inequality mirror previous evidence concerning self-reported perceptions related to this topic. Moreover, this pattern of results may suggest that conservative ideology may have specific consequences for social structures. States with a high proportion of conservatives tend to display a lack of parity between women and men—more specifically, poor well-being and health among women. While the mechanisms connecting ideology to such outcomes require more study, it is possible that political elections play a role, as conservative voters tend to prefer leaders weakly invested in disrupting the gender status quo. It is also possible that the causal arrow connecting political orientation and gender inequality runs in the opposite direction, such that living in a less gender-equal environment promotes more conservative attitudes.

The present studies present some limitations, which are crucial in envisioning key future directions. First of all, our results are correlational in nature, thus prevent us from making formal claims concerning causal directions. However, it strikes us as highly implausible that inequality at the structural level (e.g., in terms of political representation, maternal mortality, or labor-market disparities) is caused by the use (or lack thereof) of a specific Twitter hashtag. We think it is plausible that the observed correlation stems from an effect of gender inequality on #MeToo collective action. It is possible, of course, that other variables affect both gender inequality and #MeToo collective action, creating a spurious correlation between gender inequality and online activism. However, we have here tried to reduce this limitation by controlling the association between GII-S and #MeToo collective actions for several State-level factors, and the association proved robust. Further experimental studies are however needed to test additional causal processes that might have given rise to the observed correlations.

Another important limitation of our analysis lies in the fact that it is based on aggregated state-level data, which limits our ability to extrapolate our findings to individuals directly. Although our study identified the Gender Inequality Index at the state level (GII-S) as a significant predictor of participation in the #MeToo movement, even when taking general use of Twitter and state-level economic performances into account, it remains unclear whether individuals that not have the opportunity to use Internet (and Twitter) and people with lower socio-economic status have a stronger or a weaker motivation to engage in feminist collective actions. Indeed, in our studies we do not provide information about the perspectives of individuals. Consequently, survey studies that focus on individual-level analysis are essential to determine if personal socio-demographic characteristics influence attitudes toward feminist online collective actions, such as #MeToo. Despite limitations, the present studies capture a new and important picture of gender inequality in the United States. Our version of the GII not only tackles nuances of gender inequality across US states, it is also coherently associated with several well-being and health-related indicators. This demonstrates that the GII-S is a useful tool for predicting gender issues within a country. Moreover, our studies show that GII-S scores can be used to test theory-driven hypotheses in highly ecological social-media settings. Thus, the practical implications of these studies are extensive in several respects. First, by confirming the reliability of the GII-S, policymakers and advocates can use an effective tool to examine and tackle gender disparities in specific areas of the country.

Second, the outcomes of Study 1 emphasize the need to address gender inequality not just as a social justice issue but also as a means to boost in particular women's but also men's overall well-being. The awareness of the negative correlates of gender inequality for everybody can be an important element in gathering support from a larger part of the population.

Finally, Study 2 offers important insights into how gender inequality impacts collective action. The inverse relationship between GII-S scores and involvement in the #MeToo movement indicates that greater gender inequality may reduce awareness and activism regarding sexual harassment and violence against women. This is particular, relevant for activists and organizations interested in promoting gender equality, as they can consider that stronger efforts are required to mobilize people in contexts that would likely benefit most for interventions.

## Author Contributions

**Conceptualization:** Bruno Gabriel Salvador Casara, Alice Lucarini, Caterina Suitner.

**Data curation:** Bruno Gabriel Salvador Casara.

**Formal analysis:** Bruno Gabriel Salvador Casara, Alice Lucarini.

**Investigation:** Bruno Gabriel Salvador Casara, Alice Lucarini.

**Methodology:** Bruno Gabriel Salvador Casara, Alice Lucarini, Caterina Suitner.

**Project administration:** Bruno Gabriel Salvador Casara, Alice Lucarini.

**Supervision:** Eric D. Knowles, Caterina Suitner.

**Writing – original draft:** Bruno Gabriel Salvador Casara, Alice Lucarini.

**Writing – review & editing:** Bruno Gabriel Salvador Casara, Alice Lucarini, Eric D. Knowles, Caterina Suitner.

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
