## [Decision Letter · Decision Letter 0]

28 Mar 2024

PONE-D-23-22075#MeToo or “not me”?Reduced feminist collective action in US states marked by high levels of gender inequalityPLOS ONE

Dear Dr. Salvador Casara,

Thank you for submitting your manuscript to PLOS ONE. After careful consideration, we feel that it has merit but does not fully meet PLOS ONE’s publication criteria as it currently stands. Therefore, we invite you to submit a revised version of the manuscript that addresses the points raised during the review process.

We look forward to receiving your revised manuscript.

Kind regards,

Avanti Dey, PhD

Staff Editor

PLOS ONE

Journal Requirements:

2. For studies involving third-party data, we encourage authors to share any data specific to their analyses that they can legally distribute. PLOS recognizes, however, that authors may be using third-party data they do not have the rights to share. When third-party data cannot be publicly shared, authors must provide all information necessary for interested researchers to apply to gain access to the data. (https://journals.plos.org/plosone/s/data-availability#loc-acceptable-data-access-restrictions) 

4. Please ensure that you include a title page within your main document. You should list all authors and all affiliations as per our author instructions and clearly indicate the corresponding author.

6. Please upload a copy of Figure 1 and 2, to which you refer in your text on page 14. If the figure is no longer to be included as part of the submission please remove all reference to it within the text.

7. Please include your tables as part of your main manuscript and remove the individual files. Please note that supplementary tables (should remain/ be uploaded) as separate "supporting information" files

Additional Editor Comments:

The manuscript has been evaluated by two reviewers, and their comments are available below.

The reviewers have raised a number of major concerns. They feel the manuscript should outline a clearly-defined research question, and they request improvements to the reporting of methodological aspects of the study, for example, regarding the exclusion criteria and more information on how the data collection was completed. The reviewers also note concerns about the statistical analyses presented and request re-analyses be completed.

Could you please carefully revise the manuscript to address all comments raised?

Reviewers' comments:

Reviewer's Responses to Questions

**Comments to the Author**

1. Is the manuscript technically sound, and do the data support the conclusions?

Reviewer #1: Yes

Reviewer #2: No

2. Has the statistical analysis been performed appropriately and rigorously? 

Reviewer #1: Yes

Reviewer #2: I Don't Know

3. Have the authors made all data underlying the findings in their manuscript fully available?

Reviewer #1: No

Reviewer #2: Yes

4. Is the manuscript presented in an intelligible fashion and written in standard English?

Reviewer #1: Yes

Reviewer #2: Yes

5. Review Comments to the Author

Reviewer #1: The paper examines the validity of the Gender Inequality Index-S, a state-level measure of the GII in the US. To this aim, Study 1 tested how the GII-S correlates with some wellness measures among women and the Gini coefficient. Study 2 examined the link between GII-S and participation in the #MeToo movement by analyzing geo-localized messages on Twitter.

I enjoyed reading this paper, which is engaging, well-written, and generally clear. The issue has relevant implications, and the methodology is innovative.

I only have some suggestions that I hope will improve the manuscript.

First, I find the title misleading. I expect the paper to focus on #MeToo; however, the analysis of #MeToo-related Twitter messages is basically functional to demonstrate the usefulness of the GII-S in predicting participation in feminist collective action and other variables. I suggest finding a more meaningful title, considering the manuscript's focus.

On page 5 (I guess; I cannot see the page numbers), the authors should provide references for this sentence: "While massively popular, the movement has encountered a great deal of protest and criticism, especially among those who endorse a more traditional worldview regarding gender roles." Whereas I can see the point here, the following sentences refer to literature published before the #MeToo phenomenon. However, there are studies that, for instance, related sexism and attitudes toward #MeToo. Even though such studies focused on attitudes or intentions to engage rather than actual participation, they would provide some context for the sentence mentioned above (e.g., Kunst et al., 2019; Menegatti et al., 2023; Moscatelli et al., 2021

I would avoid reporting a summary of the current research’s main results at the end of the introduction (page 6).

The description of the GII reported in the Measures selection (pages 6-7, before the section on Calculation of GII-S scores) should better be moved to the theoretical introduction.

How did the authors choose that specific period for the collection of Tweets? I suggest they say something about their choice.

Concerning Study 2, the finding that gender inequality is associated with lower MeToo participation is interesting and led me to wonder whether there could be other variables that are not taken into account here. Could it be, for instance, that women's age plays a role? Could it be that states with higher gender inequality also have an older female population?

Finally, the authors should devote more effort to discussing their research's theoretical and practical implications.

Reviewer #2: Thank you for the opportunity in review this interesting work. The paper is well-written and relevant to current literature but has one important limitation not sufficiently mentioned. The structure of the paper also seems a little strange. Find below some suggestions to improve the clarity and reading of the work.

Title: The approach of use tweets should be highlight in the title. Readers could think this is a population-based study, but conclusions are based on a very specific sample.

Introduction: The text is well-written and present relevant arguments, however, is too long.

I suggest remove the 4th paragraph that is currently not adding too much.

The three paragraphs about GII could be only one with 4 points briefly: a) why measure GI; b) GII is the most used instrument; c) limitations of GII; d) between countries issues.

The paragraphs about USA are too long too. I suggest focus on Mee too directly e reduce the text drastically (only one paragraph about USA/Me too). All these statements could return in the Discussion section.

Methods:

The Methods section is a little confuse. My suggestion is the following:

a) Study 1 and 2 explanations� what they are, and which data is used with which aims (remove hypothesis).

For example: Study 1 aims proposes a GII calculates at state level and study 2 test the external validity. The sample used for study 2 was (tweets collection) …

Ethics…

b) Remove all part about limitations of GII because it is already mentioned in the introduction.

c) Calculation of GIIS scores � maintain topic

d) Convergent and Discriminant Validity Measures� maintain topic, except Table 1 description (Table 1 is results from study 1)

e) Statistical analysis � explain the description and Pearson correlation should be used. Why were these statistical methods used? Why you think correlation is a good metric for test your hypothesis?

f) Adjustments and models also should be explained in the Methods section – analysis subsection.

Results

Table 1 should be described in the Results section.

In results section, please explain what id “Hx was supported’. Reader needs to red Methods many times to remember what each Hypothesis is. Please re-write this section.

Discussion

Plos’s papers have only one Discussion section. Please, group the discussion regarding studies 1 and 2 in the general Discussion.

Topic removed from introduction could return in this section.

The paper has one big limitation that is not mentioned. Data from study 2 are not based on a population-based study but in tweets, a very biased sample. Who are the people using internet in USA? Additionally, who are the people with active voice in the USA?

For example, what is the opinion about GI or ‘Me too’ movement in very poor population, far from internet access? Feminism and women rights could be not even a priority for a family without milk or bread for the meals.

I believe, this should be strongly reinforced as a limitation and believe all conclusions should have caution in interpretation.

6. PLOS authors have the option to publish the peer review history of their article (what does this mean?). If published, this will include your full peer review and any attached files.

Reviewer #1: No

Reviewer #2: No

---

## [Author Response · Author response to Decision Letter 0]

9 May 2024

Dear Dr. Avanti Dey,

Thank you for considering our revised manuscript titled "Unveiling Gender Inequality in the US: Testing Validity of a State-Level Measure of Gender Inequality and its Relationship with Feminist Online Collective Action on Twitter" for publication in PLOS ONE. We have carefully addressed the feedback from the reviewers to improve the clarity, accuracy, and depth of our research. We believe that the manuscript has improved thanks to this round of revisions and we are grateful for the helpful comments provided to us by the editor and the reviewers.

In short, in our revision we made the following major changes:

1. We re-analyzed our data considering important factors suggested by the reviewers. This led to more precise and insightful conclusions. 

2. We expanded the discussion of our findings, better connecting them to the existing literature and emphasizing their importance in understanding both gender inequality and online activism.

Below, we address your comments and each of the editor and reviewers’ comments. We hope we have adequately addressed each of the comments but if further changes are required, we would of course be happy to oblige. We hope that the changes meet PLOS ONE's high standards.

Sincerely,

Bruno Gabriel Salvador Casara, Alice Lucarini, Eric D. Knowles, and Caterina Suitner

 

EDITOR

The manuscript has been evaluated by two reviewers, and their comments are available below.

The reviewers have raised a number of major concerns. They feel the manuscript should outline a clearly-defined research question, and they request improvements to the reporting of methodological aspects of the study, for example, regarding the exclusion criteria and more information on how the data collection was completed. The reviewers also note concerns about the statistical analyses presented and request re-analyses be completed.

Could you please carefully revise the manuscript to address all comments raised?

Response: Thank you for forwarding the insightful comments from the reviewers. We have read and carefully considered and answered each point raised. We deeply appreciate the time and expertise the reviewers have dedicated to evaluating our manuscript, and we recognize the value their feedback brings to our work. We believe that the manuscript has improved by this round of revision, thanks to these helpful comments. Below, we address all comments, providing specific details about how we have modified the manuscript. 

We hope you and the reviewers find the changes we have made to your satisfaction, and we are ready to make any further revision you consider necessary.

Sincerely,

Bruno Gabriel Salvador Casara, Alice Lucarini, Eric D. Knowles, & Caterina Suitner 

 

REVIEWER #1

POINT 1: The paper examines the validity of the Gender Inequality Index-S, a state-level measure of the GII in the US. To this aim, Study 1 tested how the GII-S correlates with some wellness measures among women and the Gini coefficient. Study 2 examined the link between GII-S and participation in the #MeToo movement by analyzing geo-localized messages on Twitter.

I enjoyed reading this paper, which is engaging, well-written, and generally clear. The issue has relevant implications, and the methodology is innovative.

Response: Thank you for taking the time to review our paper and for your positive feedback, we are glad to hear that you appreciated it and that you acknowledged the issue we addressed.

POINT 2: I only have some suggestions that I hope will improve the manuscript.

First, I find the title misleading. I expect the paper to focus on #MeToo; however, the analysis of #MeToo-related Twitter messages is basically functional to demonstrate the usefulness of the GII-S in predicting participation in feminist collective action and other variables. I suggest finding a more meaningful title, considering the manuscript's focus.

Response: We thank the Reviewer for raising this point. We understand your concerns regarding the title's alignment with the manuscript's focus, we have now changed the title of our manuscript into:

“Unveiling Gender Inequality in the US: testing validity of a state-level measure of Gender Inequality and its relationship with feminist online collective action on Twitter”

We hope that this title can better emphasize the application of the GII-S in predicting participation in feminist collective action. 

POINT 3: On page 5 (I guess; I cannot see the page numbers), the authors should provide references for this sentence: "While massively popular, the movement has encountered a great deal of protest and criticism, especially among those who endorse a more traditional worldview regarding gender roles." Whereas I can see the point here, the following sentences refer to literature published before the #MeToo phenomenon. However, there are studies that, for instance, related sexism and attitudes toward #MeToo. Even though such studies focused on attitudes or intentions to engage rather than actual participation, they would provide some context for the sentence mentioned above (e.g., Kunst et al., 2019; Menegatti et al., 2023; Moscatelli et al., 2021)

Response: Thank you for this useful feedback, we have now integrated into our introduction these references, to properly contextualize the sentence on the reception of the #MeToo movement. These changes can be found at p. 3 of the revised version of our manuscript and are reported below:

“Indeed, studies indicate that sexist beliefs and perceptions of #MeToo's effects significantly affect individuals' willingness to support the movement [28]. Moreover, Kunst and colleagues [29] reported that men's negative perceptions of #MeToo were linked to higher levels of hostile sexism and lower feminist identification, highlighting ideological rather than gender-based divisions. Additionally, internalized sexualization among women in Italy was found to negatively affects attitudes toward #MeToo and suggests that such internalization may reinforce ideologies that hinder the movement's goals [30].”

POINT 4: I would avoid reporting a summary of the current research’s main results at the end of the introduction (page 6).

Response: Following your suggestions, we have now removed the summary of main results from the introduction. The revised –and briefer– version of this paragraph can be found at p. 4 of the revised version of our manuscript and is reported below:

“Across two studies, the present research employs the Gender Inequality Index (GII) and establishes its utility in assessing state-level differences in gender disparities within the US. 

Study 1 employed population-based data retrieved from several platforms for big data storage, to create and test the validity of the state-level Gender Inequality Index (GII-S), namely an adaptation of the GII allowing within-country comparisons, which assesses state-level gender inequality across US states. In Study 2, we employed Twitter-based data to validate the GII-S externally, by examining its correlation with the prevalence of #MeToo tweets as a marker of feminist collective action and to explore the interplay between gender inequality, political ideology, and such collective actions across states.”

POINT 5: The description of the GII reported in the Measures selection (pages 6-7, before the section on Calculation of GII-S scores) should better be moved to the theoretical introduction.

Response: In line with the Reviewer’s suggestion, we have now moved this section at the end of the Introduction (p. 4 of the revised version of our manuscript)

POINT 6: How did the authors choose that specific period for the collection of Tweets? I suggest they say something about their choice.

Response: Thank you for this suggestion. We have decided to consider the timeframe from mid-January to mid-February 2019 as it aligns with the peak period of activity of Tweets on the #MeToo movement, according to an analysis of Google Trends data spanning the past five years. We have now outlined the rationale behind our decision, this information can be found at p. 8 of the revised version of our manuscript and is reported below:

“This specific timeframe was selected as matches the peak period of activity for the movement, based on Google Trends data, over the past five years (Fig 3), capturing the height of public engagement and discourse on the Google platform.”

POINT 7: Concerning Study 2, the finding that gender inequality is associated with lower MeToo participation is interesting and led me to wonder whether there could be other variables that are not taken into account here. Could it be, for instance, that women’s age plays a role? Could it be that states with higher gender inequality also have an older female population?

Response: Thank you to this very useful comment that helped us to reflect more deeply about potential confounder variables. We ran a further analysis checking whether taking into account women median age by states in 2019 would suppress the effect of GII scores on #MeToo posts frequency. While GII scores were still significantly associated with #MeToo posts frequency (r = -.46, p = .001), no evidence for an association with median women age was found (r = -.19, p = .22). Moreover, we checked whether the association between GII and #MeToo frequency was robust also when economic inequality and GDP per capita were taken into account. Once again, GII scores were significantly associated with #Metoo frequency (r = -.45, p = .003).

We have now updated the results of Study 2, in order to include this more severe test of the GII-S –#Metoo frequency association. Importantly, Importantly, we have now updated the Statistical Analyses section (p. X of the revised version of our manuscript), in order include this more severe test of the GII-S – #MeToo frequency association. (p. 9 of the revised version of our manuscript):

“Furthermore, we employed the Bayes Factor, a statistical measure used to evaluate the strength of evidence in favor of one statistical model over another, to test different models to compare the predictive strength of GII-S scores against other models. Specifically, we calculated Bayes factors, using the software JASP [49], for three linear models using #MeToo tweet frequency as dependent variable. The first (null) model included the state-level frequency of general tweets, which should trivially predict the use of any given hashtag (including #MeToo), state-level median women age, GINI, and GDP per capita. In the second (GII) model, we added state-level GII as a predictor of #MeToo tweet frequency. Finally, in the third (political) model, we added state-level political orientation, instead of GII, as a predictor of #MeToo tweets.”

Consistently, we have also updated results, in the appropriate section of the revised version of our manuscript (p. 9):

“According to the Bayes Factors analysis, the GII model was superior to both the null and political models. In particular, the observed data pattern is approximately twenty-eight times more likely to occur under GII model than null model, which can be interpreted as strong evidence [50] in favor of the model with GII as predictor.”

POINT 8: Finally, the authors should devote more effort to discussing their research’s theoretical and practical implications.

Response: Thank you for your suggestion, we have now completely reframed the discussion taking into account also the suggestions from Reviewer 2. All results from Study 1 and 2 are now more deeply discussed, and theoretical interpretation are given for all our findings. Moreover, we added a paragraph at the end of the Conclusions section, where some practical implications are more clearly outlined (p. 14):

“(…) the practical implications of these studies are extensive in several respects. First, by confirming the reliability of the GII-S, policymakers and advocates can use an effective tool to examine and tackle gender disparities in specific areas of the country. 

Second, the outcomes of Study 1 emphasize the need to address gender inequality not just as a social justice issue but also as a means to boost in particular women’s but also men’s overall well-being. The awareness of the negative correlates of gender inequality for everybody can be an important element in gathering support from a larger part of the population.

Finally, Study 2 offers important insights into how gender inequality impacts collective action. The inverse relationship between GII-S scores and involvement in the #MeToo movement indicates that greater gender inequality may reduce awareness and activism regarding sexual harassment and violence against women. This is particular, relevant for activists and organizations interested in promoting gender equality, as they can consider that stronger efforts are required to mobilize people in contexts that would likely benefit most for interventions.”

REVIEWER #2

POINT 1: Thank you for the opportunity in review this interesting work. The paper is well-written and relevant to current literature but has one important limitation not sufficiently mentioned. The structure of the paper also seems a little strange. Find below some suggestions to improve the clarity and reading of the work.

Title: The approach of use tweets should be highlight in the title. Readers could think this is a population-based study, but conclusions are based on a very specific sample.

Response: We sincerely appreciate the time you invested in reviewing our paper and we thank you for recognizing the significance of the issue we addressed. 

Concerning the title, we align with your comment and we acknowledge that, in its current form, the title might be misleading, a point that was also raised by Reviewer 1. We have now changed the title, and we hope it is now in line with both your feedback and that of Reviewer 1. As reported above, the new title is:

“Unveiling Gender Inequality in the US: testing validity of a state-level measure of Gender Inequality and its relationship with feminist online collective action on Twitter”

With this changes, we have tried to better outline the online context of our work, by mentioning Twitter in the title. We did, however, chose not to directly emphasize the use of Twitter-based data in the title because, despite Study 2 is not a population-based study (being based on Tweets frequency), Study 1 is. Indeed, to calculate GII-S scores we relied on data from several platforms for big data storage, such as Gallup and the United State Census Bureau, whose data are population-based. Currently, the new title of our manuscript does meet your suggestion to underline the online component of this work, as we refer to online feminist collective action on Twitter, while also aligning with Reviewer’s 1 comment, who suggested to put more focus on the GII-S and on its predictive role of feminist collective action. We hope that this solution aligns with your request. 

POINT 2: Introduction: The text is well-written and present relevant arguments, however, is too long.

I suggest remove the 4th paragraph that is currently not adding too much.

Response: In line with your suggestions, we have now removed the 4th paragraph from the revised version of our manuscript.

POINT 3: The three paragraphs about GII could be only one with 4 points briefly: a) why measure GI; b) GII is the most used instrument; c) limitations of GII; d) between countries issues.

The paragraphs about USA are too long too. I suggest focus on Mee too directly e reduce the text drastically (only one paragraph about USA/Me too). All these statements could return in the Discussion section.

Response: Thank you for your constructive feedback. In response to your comments, we have streamlined the contents discussed in the Introduction, to ensure it remains concise and focused on the core aspects of the Gender Inequality Index (GII) and the #MeToo movement's impact in the USA. Here's the revised version that incorporates your suggestions (p. 2):

“Fighting gender inequality requires measuring it, and specific instruments have been developed to assess disparities between men and women. One of the most important measures of gender inequality is the Gender Inequality Index (GII; [26]), a measure developed to assess gender inequality at country level, bu

---

## [Decision Letter · Decision Letter 1]

3 Jun 2024

PONE-D-23-22075R1Unveiling Gender Inequality in the US: testing validity of a state-level measure of Gender Inequality and its relationship with feminist online collective action on TwitterPLOS ONE

Dear Dr. Salvador Casara,

Thank you for submitting your manuscript to PLOS ONE. After careful consideration, we feel that it has merit but does not fully meet PLOS ONE’s publication criteria as it currently stands. Therefore, we invite you to submit a revised version of the manuscript that addresses the (minor) points raised during the review process. Due to a change in the Editor, I, as a guest Editor, asked the opinion of a third reviewer. The reviewer is positive about the manuscript and provided important suggestions, which I invite you to follow. Please submit your revised manuscript by Jul 18 2024 11:59PM. If you will need more time than this to complete your revisions, please reply to this message or contact the journal office at plosone@plos.org. Please include the following items when submitting your revised manuscript:A rebuttal letter that responds to each point raised by the academic editor and reviewer(s). You should upload this letter as a separate file labeled 'Response to Reviewers'.A marked-up copy of your manuscript that highlights changes made to the original version. You should upload this as a separate file labeled 'Revised Manuscript with Track Changes'.An unmarked version of your revised paper without tracked changes. You should upload this as a separate file labeled 'Manuscript'.If applicable, we recommend that you deposit your laboratory protocols in protocols.io to enhance the reproducibility of your results. Protocols.io assigns your protocol its own identifier (DOI) so that it can be cited independently in the future. For instructions see: https://journals.plos.org/plosone/s/submission-guidelines#loc-laboratory-protocols. Additionally, PLOS ONE offers an option for publishing peer-reviewed Lab Protocol articles, which describe protocols hosted on protocols.io. Read more information on sharing protocols at https://plos.org/protocols?utm_medium=editorial-email&utm_source=authorletters&utm_campaign=protocols.

We look forward to receiving your revised manuscript.

Kind regards,

Silvia Moscatelli

Guest Editor

PLOS ONE

Journal Requirements:

Reviewers' comments:

Reviewer's Responses to Questions

**Comments to the Author**

1. If the authors have adequately addressed your comments raised in a previous round of review and you feel that this manuscript is now acceptable for publication, you may indicate that here to bypass the “Comments to the Author” section, enter your conflict of interest statement in the “Confidential to Editor” section, and submit your "Accept" recommendation.

Reviewer #3: All comments have been addressed

2. Is the manuscript technically sound, and do the data support the conclusions?

Reviewer #3: Yes

3. Has the statistical analysis been performed appropriately and rigorously? 

Reviewer #3: Yes

4. Have the authors made all data underlying the findings in their manuscript fully available?

Reviewer #3: Yes

5. Is the manuscript presented in an intelligible fashion and written in standard English?

Reviewer #3: Yes

6. Review Comments to the Author

Reviewer #3: Thank you for the opportunity to revise the manuscript entitled “Unveiling Gender Inequality in the US: testing validity of a state-level measure of Gender Inequality and its relationship with feminist online collective action on Twitter”. This is a revised version of the manuscript and I think that the authors adequately addressed the comments raised by the Reviewers. The results of the two studies contribute to the understanding of gender inequality and its social and psychological correlates in the US, providing useful insights for practical implications toward greater equality. The comments that follow are minor and they point to the overall improvement of the manuscript:

1. I suggest revisiting the occasional use of casual language, whether implicit or explicit, given the adopted correlational design. Precision is critical in eliminating any potential for misunderstanding.

2. My second comment regards the lack of significant association between the GII-S and GINI coefficient. The authors discuss that “economic and gender disparities represent distinct aspects of social inequality”. However, there is literature suggesting that economic and gender inequality are related to each other (see Moreno-Bella et al., 2023 for a recent paper). Usually, societies characterized by higher levels of economic inequality are also characterized by higher levels of gender economic inequality (i.e., gender pay gap). In economically unequal societies, women are usually disadvantaged compared to men. Therefore, I would suggest being more cautious in interpreting this lack of significant association. This could be due to the specific context the authors focused on (i.e., the US) or the specific measure they used (i.e., if I understood correctly, the GII-S does not account for gender economic inequality).

3. Along the same lines as my previous comment, the authors discuss the negative association between the GII-S and MeToo hashtags from the perspective of the system justification theory. This could be a possibility, but further evidence is needed before concluding that e.g., GII-S undermines feminist collective action through higher perceptions of gender inequality as legitimate. Again, the authors should be more cautious in interpreting these results and propose additional explanations. For example, given the specific inequality dimensions accounted by the GII-S, could be that this negative association is due to the lack of resources, empowerment, and group efficacy beliefs? Relatedly, this could be due to the specific measure the authors used for collective action (i.e., MeToo hashtags) and other measures of collective action engagement (i.e., demonstrations, strikes) should be examined.

7. PLOS authors have the option to publish the peer review history of their article (what does this mean?). If published, this will include your full peer review and any attached files.

Reviewer #3: No

---

## [Author Response · Author response to Decision Letter 1]

5 Jun 2024

Reviewer

Thank you for the opportunity to revise the manuscript entitled “Unveiling Gender Inequality in the US: testing validity of a state-level measure of Gender Inequality and its relationship with feminist online collective action on Twitter”. This is a revised version of the manuscript and I think that the authors adequately addressed the comments raised by the Reviewers. The results of the two studies contribute to the understanding of gender inequality and its social and psychological correlates in the US, providing useful insights for practical implications toward greater equality. The comments that follow are minor and they point to the overall improvement of the manuscript.

Response: We are pleased to hear that you believe we have adequately addressed the comments raised by the Reviewer, and we thank you for your constructive feedback.

POINT 1: I suggest revisiting the occasional use of casual language, whether implicit or explicit, given the adopted correlational design. Precision is critical in eliminating any potential for misunderstanding.

Response: Thank you for your valuable feedback. We have carefully reviewed the manuscript and identified instances where language could be interpreted as implying causality. These instances have been revised to ensure clarity and precision, reflecting the correlational nature of our study. We have replaced any language that may have suggested causal relationships with terminology that accurately describes associations and correlations. For example, we are now avoiding the term “predicted” and we are using the term “associated”.

POINT 2: My second comment regards the lack of significant association between the GII-S and GINI coefficient. The authors discuss that “economic and gender disparities represent distinct aspects of social inequality”. However, there is literature suggesting that economic and gender inequality are related to each other (see Moreno-Bella et al., 2023 for a recent paper). Usually, societies characterized by higher levels of economic inequality are also characterized by higher levels of gender economic inequality (i.e., gender pay gap). In economically unequal societies, women are usually disadvantaged compared to men. Therefore, I would suggest being more cautious in interpreting this lack of significant association. This could be due to the specific context the authors focused on (i.e., the US) or the specific measure they used (i.e., if I understood correctly, the GII-S does not account for gender economic inequality).

Response: Thank you for your valuable feedback. We have carefully considered your comment regarding the lack of significant association between the GII and the Gini coefficient. While we see the consistency of our result with other empirical evidence showing that the relationship between GII and GINI is not necessarily positive nor negative, we completely agree with the point raised by the reviewer. Thus, we added the following paragraph (p. 12):

“Our findings indicate that GII-S scores are not significantly correlated with economic inequality, as measured by the state-level Gini coefficient. Our findings align with the idea that economic inequality and gender inequality are distinct phenomena. For example, reducing disparities between men and women does not automatically imply that economic differences between classes will follow the same trend. Moreover, coherently with our results, from the 1990 Country-level gender inequality measured with the GII is decreasing in the World and in many countries, USA included [62]. Differently, economic inequality appears to be more stable over time and it increased in the USA during the same time period [63]. However, it is important to notice that there are social issues representing both facets of gender and economic inequality, such as the gender wage gap. Furthermore, our data shows that the Gini coefficient is associated with women’s financial well-being (but not men’s), and there is previous research reporting a link between economic and gender inequality [64]. Additionally, the distinctiveness between the two issues may be particularly salient because of the specific aspects of gender inequality that the GII-S measures, which do not directly account for gender economic inequality. Therefore, we advise caution in interpreting the lack of a significant association between these two measures.”

POINT 3. Along the same lines as my previous comment, the authors discuss the negative association between the GII-S and MeToo hashtags from the perspective of the system justification theory. This could be a possibility, but further evidence is needed before concluding that e.g., GII-S undermines feminist collective action through higher perceptions of gender inequality as legitimate. Again, the authors should be more cautious in interpreting these results and propose additional explanations. For example, given the specific inequality dimensions accounted by the GII-S, could be that this negative association is due to the lack of resources, empowerment, and group efficacy beliefs? Relatedly, this could be due to the specific measure the authors used for collective action (i.e., MeToo hashtags) and other measures of collective action engagement (i.e., demonstrations, strikes) should be examined.

Response: Thank you for your comment which helps us to provide multiple interpretations for our results. Now, we are suggesting that our results are coherent with the SJT account, but we also consider other theoretical perspectives coherent with our results. We add this in the discussion section (p. 13):

“Results showed that states with high levels of gender inequality tended to produce fewer #MeToo tweets than did states low in gender inequality. One possible interpretation of these findings can be framed in the theoretical account of the System Justification Theory, in line with our preregistered hypothesis (SJT; [65]). According to SJT, people in macro-level contexts characterized by a long history of patriarchy and adherence to traditional sex roles will be motivated to justify gender inequality—a process that likely reduces interest in feminist collective action. In this way, inequality’s very existence may further resistance to change. However, it is important to highlight that, while our results are consistent with an interpretation based on System Justification Theory (SJT), they do not conclusively prove it. Other interpretations are also plausible. For instance, according to the Social Identity Model of System Attitudes (SIMSA; [66]), the association between higher GII scores and lower frequency of #MeToo hashtags might be due to accuracy motives. These motives could lead individuals to passively perceive and acknowledge the status quo without actively defending or maintaining the system. Additionally, as the GII measures objective structural inequalities between men and women, it is possible that in States with higher levels of gender inequality, there is a lack of resources and opportunities to participate in the #MeToo movement. Resource scarcity may also represent an obstacle in the expression of important psychological antecedents of collective action, such as feelings of empowerment and group efficacy beliefs [67]. It is also worth considering that our measure of collective action engagement was specific to the #MeToo hashtags. Future research should examine other forms of collective action, such as participation in demonstrations and strikes. By doing so, researchers might gain a more comprehensive understanding of the dynamics at play, while also providing a more direct test of their possible explanation.”________________________________________

---

## [Editor Report · Decision Letter 2]

12 Jun 2024

Unveiling Gender Inequality in the US: testing validity of a state-level measure of Gender Inequality and its relationship with feminist online collective action on Twitter

PONE-D-23-22075R2

Dear Dr. Salvador Casara,

We’re pleased to inform you that your manuscript has been judged scientifically suitable for publication and will be formally accepted for publication once it meets all outstanding technical requirements.

Kind regards,

Silvia Moscatelli

Guest Editor

PLOS ONE
---

## [Editor Report · Acceptance letter]

21 Jun 2024

PONE-D-23-22075R2 

PLOS ONE

Dear Dr. Salvador Casara, 

I'm pleased to inform you that your manuscript has been deemed suitable for publication in PLOS ONE. Congratulations! Your manuscript is now being handed over to our production team.

Kind regards, 

on behalf of

Dr. Silvia Moscatelli 

Guest Editor

PLOS ONE